# Influences of Climate Change and Land Use Change on the Habitat Suitability of Bharal in the Sanjiangyuan District, China

**DOI:** 10.3390/ijerph192417082

**Published:** 2022-12-19

**Authors:** Shengwang Bao, Fan Yang

**Affiliations:** School of Economics and Management, Zhejiang Ocean University, Zhoushan 316022, China

**Keywords:** bharal, ecological corridors, climate change, habitat suitability, Sanjiangyuan District

## Abstract

One of the biggest dangers to the degradation of biodiversity worldwide is climate change. Its oscillations in the future could result in potential alterations to species populations and habitat structure. With Sanjiangyuan District as the study site, an uncrewed aerial vehicle (UAV) was utilized to investigate the number and location of the bharal (*Pseudois nayaur*). The Maximum Entropy model and the Minimum Cumulative Resistance model (MaxEnt-MCR) were coupled to simulate the distribution of wildlife. On this basis, the future geographical distribution of bharal under different climate scenarios was simulated, and the ecological corridor and habitat centroid of bharal were revealed. The results showed that the suitable area of the bharal habitat was 4669 km^2^, which was mainly concentrated in the Maduo, Qumalai, and Gonghe counties. The potential distribution of the species under different future climate scenarios had a decreasing trend. Under the SSP-245 scenario, the habitat area of bharal in 2030 and 2050 decreased by 25.68 and 44.61% compared with the present situation and cumulatively decreased by 1199 and 2083 km^2^, respectively. Under the SSP-585 scenario, the habitat area of bharal in 2030 and 2050 decreased by 27.5 and 48.44%, with a total reduction of 1284 and 2262 km^2^, respectively. Furthermore, a complete loss of habitat was predicted in Gonghe County by 2050. In addition, it was observed that the landscape structure in Sanjiangyuan District would be more fragmented and complex. The continued climate change will seriously affect the habitat distribution of this species. Therefore, preventive measures, such as protecting habitat areas and establishing ecological corridors for bharal, should be implemented in the Sanjiangyuan District. Such measures should not focus solely on the potential degradation but should also be extended to include potential distribution areas for future migration.

## 1. Introduction

Climate is an imperative factor affecting biodiversity [1]. Climate change has already led to the destruction of many wildlife habitats and diversity [2,3]. The inability to adapt quickly to changes in climatic conditions has led to the extinction of wildlife [4]. Regarding whether humans satisfy the emission reduction targets, there is a risk of a precipitous decline in global biodiversity [5,6]. The Sanjiangyuan District is a typical alpine ecosystem with a high degree of endemism and a wide range of specificities [7]. With the severity of grass–livestock conflicts [8,9,10], it is crucial to explore the influence of environmental factors on the spatial dynamics and evolution of habitats [11]. The bharal (*Pseudois nayaur*) is a kind of sheep that mainly lives in China on the Qinghai–Tibet Plateau, Gongga Mountains, and Helan Mountains [12,13]. Bharal is a large plateau-dwelling herbivore with high coverage of grass and was listed as non-threatened by the IUCN in 2014 [12]. With the degradation of plateau ecosystems [8], the habitats of large herbivores are under a serious threat [14,15].

Habitat refers to the critical resources and subsistence environments for species and communities to survive [2,3]. Human activities, dominated by excessive grazing and extraction practices, have directly led to environmental deterioration, thus posing a great threat to the habitat of species [16]. The loss of habitat also makes species migration difficult and threatens the conservation and development of biodiversity [17,18]. Species that are weakly adapted and sensitive to environmental responses are likely to become extinct [19,20]. The ecosystems in the Sanjiangyuan District are very fragile, showing extreme sensitivity to climate fluctuations. The habitats of bharal in this district, a typical representative of large herbivores [21], are seriously threatened. Therefore, it is necessary to investigate the number and population distribution of bharal [13,22] and analyze the impact of climate change on the bharal habitat from the perspective of protecting the species’ habitat and diversity distribution.

At present, the line intercept transect method is mainly adopted for the survey of wildlife distribution, number, and locations [23]. However, in practice, the line intercept transect method is prone to duplicate counts due to animal escapes and the influence of environment and terrain on species [24]. The emergence of uncrewed aerial vehicles (UAVs) may improve this condition. Uncrewed aerial vehicles (UAVs) show great advantages in the survey of bharal points because they are not affected by terrain and have little disturbance, which provides a new means for wildlife surveys [25]. Precise bharal points are beneficial in improving the accuracy of species distribution modeling. Species distribution models (SDMs) estimate the ecological niche of a species from known species occurrence points and environmental data via specified algorithms and ultimately present the species’ preference for habitat use probabilities [26]. It is possible to simulate the spatial distribution of species and the degree of response to the environment by using SDMs [27]. Of these models, the Maximum Entropy model (MaxEnt model) has optimal performance and high accuracy even with small-scale samples [28]. Therefore, it was decided to investigate the field distribution of bharal by UAVs combined with the historical presence of occurrence sites and employ the MaxEnt model to clarify habitat characteristics and potential geographic distribution [29]. In this way, the habitat change trends of bharal under different future climate change contexts can be accurately analyzed and evaluated from the perspective of spatial dynamics [30]. With the dominant factors affecting the habitat distribution of the species being revealed [31,32], proactive conservation strategies can be adopted to mitigate and reduce the adverse effects of future climate change on the species [33,34].

In this study, precise distribution points of bharal in the Sanjiangyuan District were collected by using UAVs to model the geographic distribution of bharal accurately [25] and to study contemporary climate changes in the habitat distribution. First, the MaxEnt model with contemporary climate was employed to infer the spatial pattern of distribution, and the future distribution of bharal under different future climate scenarios was predicted. Second, according to the response to environmental factors, the future land use and land cover changes (LUCC) under different climate scenarios were simulated to improve the accuracy of simulating bharal distributions. Finally, a new coupled MaxEnt model and Minimum Cumulative Resistance model (MaxEnt-MCR model) was proposed to analyze the ecological resistance in the Sanjiangyuan District and disclose the ecological corridors to show the paths of survival and migration, which is important for species diversity conservation in the Sanjiangyuan District.

## 2. Materials and Methods

### 2.1. Study Area

The Sanjiangyuan District (31°39′~36°12′ N, 89°45′~102°23′ E) is located in the south of Qinghai Province in China, with an average altitude of 3500 to 4800 m (Figure 1), which is the hinterland and main body of the Tibetan Plateau, dominated by mountainous terrain. The climate of the region is typical of a continental plateau, with alternating hot and cold seasons and distinct wet and dry seasons. The mountain range is crisscrossed by glaciers, which are one of the most concentrated glaciers in China and have the highest altitude and richest wetland types in the world. There are 69 species of national key protected animals, such as the bharal and Tibetan gazelle (*Procapra picticaudata*) at the national level. In recent decades, the ecological environment of the entire Tibetan Plateau has been notably deteriorating due to climate change and severe soil erosion, and the threatened biological species account for more than 20% of the total category.

### 2.2. Data Sources

#### 2.2.1. Species Occurrence Point

The Global Biodiversity Information Facility (GBIF) is a relatively complete species diversity database that records the geographic locations of bharal occurrence sites on a global scale. As the GBIF database shows, bharal is mainly distributed in locations of Central Asia, such as India, Nepal, Bhutan, and western China. Within China, bharal are mainly distributed in Tibet, the Helan Mountains in Sichuan, and the Sanjiangyuan District in Qinghai. Therefore, the occurrence sites in the Sanjiangyuan District were selected to enhance the completeness of bharal in the study area.

This database was utilized, and corrections were made for some areas of the Sanjiangyuan District as follows: during interviews with herders, it was reported that bharal were mainly distributed in the area around Huashixia in Maduo County, but few traces were found in the resting place. Therefore, an extensive survey of Maduo County was performed using UAV, which is an effective supplement to the GBIF database and improves the completeness of bharal occurrence sites in the Sanjiangyuan District. In this study, the systematic sampling method was adopted to lay out 14 survey sample strips. At the same time, the survey sample strips were made to contain as many geographical information elements as possible, and the accessibility of UAV landing points was considered. The UAV survey sample strips were designed by referring to the 2011 National Forestry Administration’s “National Second Terrestrial Wildlife Resources Survey Technical Regulations”, considering topography, LUCC, and vegetation type. Under the guidance of relevant wildlife research experts, a systematic sampling method was adopted to distribute the survey sample strips evenly. All the aerial photography of bharal (*Pseudois nayaur*) was taken by taking 14 sample strips, and all photography was captured between 7:00 and 11:00 from 9 to 18 April 2017. The aerial photography was taken by two UAVs. The first one was an electric fixed-wing UAV developed by the Institute of Mountain Hazards and Environment, CAS., The second one was an F1000 electric fixed-wing UAV from Shenzhen Feima Robotics Co., Ltd. The specific parameters of both UAVs are shown in Table 1. Each sample strip was shot by a single UAV to avoid the impact of repeated interpretation caused by the movement of wild animals. Two high-performance workstations for stitching and five computers were employed for manual visual interpretation. Pix4Dmapper and LiMapper were utilized for image stitching, and ArcGIS was used for visual interpretation. A total of 203 bharal occurrence sites were acquired based on the research of Guo [25]. The bharal distribution points were collected from the GBIF and UAV surveys. Points smaller than 100 m were discarded to avoid the impact of autocorrelation caused by too-dense points in the species distribution modeling process, and one was kept randomly. In total, 222 bharal occurrence sites were reserved.

#### 2.2.2. Environmental Factors

The selected data is classified into natural geography, human disturbance, and bioclimate factors regarding their attributes. The natural geographic factors include the Digital Elevation Model (DEM, https://www.gscloud.cn/ accessed on 22 August 2022), slope (SLO), aspect (ASP), land use and land cover (LUCC, https://www.resdc.cn/ accessed on 23 August 2022), distance to water (DW), distance to high coverage of grassland (DH), net primary productivity (NPP, http://www.ntsg.umt.edu/project/modis/mod17.php accessed on 25 August 2022), and normalized difference vegetation index (NDVI, https://www.resdc.cn/ accessed on 23 August 2022). Human disturbance factors are reflected as distance to the road (DW) from the National Directory of Geographic Information Resources (https://www.webmap.cn/ accessed on 27 August 2022). Bioclimate factors include contemporary bioclimate (1970–2000) and future bioclimate (SSP-245; SSP-585); factors of 2030 were taken as a representative of 2021–2040, and 2050 as a representative of 2041–2060. Nineteen variables (abbreviated as Bio1-Bio19) are included in each climate scenario, all of which were from the ACCESS-CM2 site. SLO and ASP were extracted from DEM, and DH and DW were extracted from LUCC. The workflow chat is shown in Figure 2.

Future bioclimates under different scenarios based on CMIP6 have coupled with the Shared Social-economic Pathways (SSPs) and Representative Concentration Pathways (RCPs) [35]. SSP-245 upgrades the RCP4.5 scenario via SSP2 (medium-forcing scenario), representing the medium pathway for future emissions. SSP-585 upgrades the RCP8.5 scenario via SSP5 (high-forcing scenario). The two scenarios can effectively simulate and reflect future climate fluctuations and changes under different human activity intensities [36].

Given the problem of multicollinearity among different variables, all environmental factors were used to fabricate the initial model, and the Pearson method was adopted to analyze the correlations between climatic and environmental factors (Figure 3). Overall autocorrelation analysis was performed on all variables, and it was pre-modeled by using the Maximum Entropy model. The model was repeatedly established ten times, and the test sample was set to 25% with a convergence threshold of 0.00001. The contribution of variables was tested using the jackknife method. The variables with |r|<0.8 and high relative contribution rates were selected to work in the model prediction. Eventually, a total of 13 factors were chosen for the species distribution modelling.

### 2.3. CA–Markov Model

Markov models are widely used in land use and land cover prediction, for they have no after-effect and can predict the state of things in the future [37]. The equation is expressed as follows:(1)$$S_{\left(t+1\right)}=S_{t}\times P_{ij}$$

$S_{t}$ is the land use state at the moment t, $S_{\left(t+1\right)}$ is the land use state at the moment, and t+1 and $P_{ij}$ are the land use transfer matrix. The CA–Markov model can predict the future land use state both spatially and quantitatively. Hence, LUCC, DW, and DH in different scenarios were predicted by using modified Markov prediction to simulate the future land use changes under the future climate fluctuations, contributing to accurate simulation of bharal distribution.

### 2.4. Coupled Maximum Entropy Model and Minimum Cumulative Resistance Model (MaxEnt-MCR Model)

The Maximum Entropy model (MaxEnt model) is a spatial species distribution of species at geographic scales under the maximum entropy theory, with a high degree of confidence relative to other species distribution models [38]. The area under the curve (AUC) was calculated to evaluate the accuracy of MaxEnt model prediction results. In addition, the contribution of variables, ranking importance, and the jackknife cut test are useful for reflecting the impact of variables in model construction [39].

According to the description in the Fifth Assessment Report of the IPCC [40], the suitable zones were classified into four levels, namely unsuitable area (0–0.2), marginally suitable area (0.2–0.5), moderately suitable habitat area (0.5–0.7), and most suitable habitat area (0.7–1).

Ecological source sites are potential distribution concentration areas and the most suitable habitat centers for species [41]. The Minimum Cumulative Resistance model (MCR model) was used to establish the minimum cumulative resistance surface, and the work between ecological source sites was calculated to generate the optimal path for biological migration to avoid external interference [42]. The equation of the MCR model is as follows:(2)$$MCR=f_{min}\overset{i=m}{\underset{j=n}{\sum }}D_{ij}R_{i}$$
where $D_{ij}$ denotes the spatial distance from source j to source i; $R_{i}$ denotes the resistance value of spatial cell i; f represents the minimum cumulative resistance at any point as a function of the distance to the sources.

The MaxEnt-MCR model can identify important ecological source sites regarding spatial aggregation extent and area size according to MaxEnt simulation results. Then, it can construct an ecological network using the MCR model by combining the resistance surface system. During the construction, the threshold of each environmental factor and the resistance value were determined by the MaxEnt Model. The resistance surface of each factor was constructed and superimposed regarding the contribution of each environmental factor. Cost paths were used to reveal ecological corridors interacting with different ecological source sites.

### 2.5. Landscape Pattern Index

Fragstats (V4.2) was used to analyze the characteristics of the habitat landscape pattern [43]. Six indices were chosen and calculated, namely the number of patches (NP), spread (CONTAG), patch density (PD), division index (DIVISION), landscape shape index (LSI), and Shannon diversity index (SHEI) of the habitat landscape in the study area. NP indicates the number of various types of patches in the habitat landscape; CONTAG describes the degree of patch types clustering in the habitat landscape (the higher the value, the lower the fragmentation); PD indicates the degree of fragmentation of the habitat landscape; DIVISION reflects the degree of fragmentation of the landscape (the higher the degree of separation, the higher the degree of dispersion in [0, 1]); LSI reflects the complexity of the habitat; SHEI measures the complexity of the landscape structure (the higher the value, the more diverse of the patch types).

## 3. Results

### 3.1. Revealing Main Factors

The prediction results showed that the value of average AUC after ten replications was 0.995 ± 0.004 (Mean ± SD), which indicated that the prediction results by the MaxEnt model had high accuracy according to Phillips’ evaluation of the model itself.

The main environmental factors affecting the current distribution of bharal habitat suitability areas were analyzed based on the relative contribution of environmental factors, and the most significant ones were selected, including Precipitation of wettest quarter-Bio16 (32.4%), Precipitation seasonality-Bio15 (21.4%), Isothermality-Bio3 (13.1%), DR (12.7%), DH (10.9%), and LUCC (5.6%).

As the jackknife test indicated (Figure 4), Bio3, Bio16, Bio15, DH, DR, and LUCC had high AUC values, with Bio3 generating the greatest gain, which suggested that Bio3 performed the best when coupled with the distribution pattern of suitable habitat for bharal. When the model was constructed using Bio7, DW, and Bio19 alone, it had the least effect on the distribution. Combined with the contribution of each environmental factor, these six environmental factors above were deemed as the dominant environmental factors in determining the species distribution of bharal.

According to the results of the MaxEnt model, the influence thresholds on the species distribution of bharal were analyzed and were classified into five categories for each influence interval with values of 10, 20, 30, 40, and 50. By discussing the influence intervals and setting the range of resistance values, the minimum cumulative resistance surface for the Sanjiangyuan District was established (Table 2). In addition, the relative contribution of each environmental factor to the suitable habitat by the output of the MaxEnt model was taken as the weight of the resistance factor in MCR.

### 3.2. Current Habitat Analysis of Bharal

Under the modern climate, the total area of habitat in the Sanjiangyuan District is 4669 km^2^ (Figure 5), of which 2668 km^2^ is marginally suitable, accounting for 57.1% of the total area. The moderately suitable habitat area accounts for 16.4%, and the most suitable habitat area of 1237 km^2^ accounts for 26.5%. The most suitable habitat area is concentrated in Maduo County (33°50′~35°40′ N, 96°50′~99°20′ E), Qumalai County (92°56′~97°35′ E, 33°36′~35°40′ N), and Gonghe County (99°~101.5° E, 35.5°~37.2° N). The suitable habitat area of bharal in Maduo County covers 3837 km^2^, concentrated in the northern part of the county, surrounding Zaling Lake and the source of the Yellow River, with an average altitude of 4500 m. Additionally, a small number of suitable areas are concentrated in the northwestern part of Gonghe County, and the area of the bharal habitat under jurisdiction is 487 km^2^. The suitable habitat area in Qumalai County is 326 km^2^. In general, the spatial structure of suitable habitat shows aggregation and singleness.

### 3.3. Future Evolution of Suitable Habitat and Changes in Landscape Patterns

#### 3.3.1. Land Use and Land Cover Changes in Different Scenarios

The land transfer matrix of both SSP-245 and SSP-585 is shown in Figure A1. The simulation of LUCC under SSP-245 and SSP-585 in the future 2030 and 2050 is shown in Figure A2.

#### 3.3.2. Spatial–Temporal Dynamics of Suitable Habitat

Future climate data was collected to simulate the potential distribution habitat of bharal. The environmental factors of DH and DW were derived from simulations of LUCC under different climate features based on the SSP-245 and SSP-585. The methods and simulation results of LUCC can be found in Appendix A.

The predicted results were used as environmental factors under SSP-245 and SSP-585. The future area of the suitable zone of bharal is shown in Table 3. By 2030, the total area of suitable habitat in the Sanjiangyuan District will be 3470 km^2^ under the SSP-245, 25.6% less than the current situation. Under SSP-585, the suitable habitat was reduced by 44.5%. By 2050, the total habitat area of bharal will be 2407 km^2^ under SSP-585, which is still decreasing. The areas of habitat in different climatic scenarios are shown in Table 3. A rapid decrease in suitable habitat areas was indicated under SSP-585 than in SSP-245, with marginally suitable habitat areas, moderately suitable habitat areas, and most suitable habitat areas decreasing rapidly. In general, the area of suitable zone will keep decreasing in the future, and it will decrease even more rapidly under SSP-585 than under SSP-245 due to the climate change caused by human activities.

In terms of the spatial pattern (Figure 6), the habitat suitable in Gonghe County is gradually lost, and 2 km^2^ (SSP-245) and 1 km^2^ (SSP-585) are left in 2050 under SSP-245 and SSP-585, respectively, which is basically in the state of a complete loss. Moreover, the suitable habitat of bharal is mainly reduced within the marginally suitable habitat under SSP-245, and the habitat will decline drastically by 57.2% in 2050 under SSP-585. The suitable habitat has been basically lost in Qumalai County.

### 3.4. Evolution of Landscape Pattern Index

The landscape pattern indices of suitable habitats are shown in Figure 7. Under different climatic scenarios, PD, LSI, DIVISION, and SHEI indices have decreasing trends, while the index of CONTAG has an increasing trend, which indicates the fragmentation, complexity, and homogenization of suitable habitats. The decrease in NP clearly reflects the shrinkage. Under SSP-245, the landscape pattern index decreases relatively slowly, and the fragmentation and homogenization of potential habitat increases, while under SSP-585, the landscape pattern index changes greatly in the first ten years and slightly in the last 40 years. The extreme climate caused by human activities will have a major impact on the suitable habitat of the bharal.

## 4. Discussion

### 4.1. Transition of the Suitable Habitat Distribution for Bharal

Future climate fluctuations have a considerable impact on the spatial distribution of habitats [6]. Excessive human emissions will lead to climate and environmental degradation, thereby causing species habitat loss [16,44]. Indeed, human activities disturb species distribution and diversity in many ways [45]. Global climate and environment are strongly associated, and many countries have now taken measures to restrain the impact on the environment. For example, the Paris Agreement [46] is a joint human effort to manage the global climate. The potential habitat center of the bharal describes the spatial distribution of geographical things and can reflect the characteristics of geographical elements as a response to global climate change under future climate scenarios [47]. Under the influence of two shared socioeconomic pathways, the potential habitat center of mass for bharal shifted toward lower latitudes. Under SSP-245, the center shifts to the southwest in 2030, with an average elevation drop of 26 m. In 2050, the center shifts to the southeast from 2030, with an elevation drop of 40 m. The potential habitat center of bharal shifts to the southwest from 2030 to 2050 under SSP-585, with an average elevation drop of 197 m (Figure 8). The change of the habitat center indicates the direction of potential distribution and foreshadows the future migration path of bharal. This may also be one of the reasons why the Yushu area has no suitable habitats where the GBIF points can be found.

Species occurrence points represent the geographic locations where species have appeared [48]. Unlike the fixed nature of plants, bharal have migratory and mobile characteristics. Since Hodgson’s survey in 1833 [49], bharal have migrated over longer distances. Habitats represent areas suitable for species distribution and areas where species preferentially select [50]. Therefore, the occurrence point does not necessarily mean that it is a suitable habitat zone and may cause the phenomenon that the species occurrence point does not overlap with the suitable habitat area [51]. In addition, the threshold of the MaxEnt model affects the potential habitat distribution of bharal as well. Generally speaking, the larger the threshold value is, the smaller the potential habitat distribution area is. At present, there are several methods to select the threshold value: Frist, defining the threshold value according to ecological principles or expert experience; Second, making the assessment method related to the threshold value to obtain the one with the best model evaluation result; Third, calculating the probability threshold value by the ROC curve; Fourth, the lowest existence threshold value, with its lowest probability as the classification threshold value [52]. In this study, referring to the threshold selection of bharal and other similar species habitat distribution studies, combined with the characteristics of the Sanjiangyuan District and the fifth report of IPCC, 0.2 was selected as the minimum threshold that can effectively distinguish the category of bharal habitat suitable areas. This may be also one of the reasons why there seems to be no suitable rock sheep habitat areas in the Yushu area.

### 4.2. Ecological Corridor and Habitat Optimization Recommendations

Migration is the main strategy for wildlife adaptation to global warming, during which most wild species face a habitat loss crisis [53]. Currently, natural reserves have limited contributions to the conservation of species. Hence, identifying species’ habitats accurately and establishing ecological corridors will be conducive to enhancing the connectivity among their habitats and reducing habitat loss and isolation, which is the best approach to protect the sustainable survival of species [54]. The established resistance surface in the Sanjiangyuan District by the MaxEnt-MCR model is shown in Figure 9. Ecological source sites of bharal traverse in the Qumalai, Maduo, and Gonghe counties are concentrated in Maduo County, with a small number of distributions in the Gonghe and Qumalai counties. Gonghe County lies near Qinghai Lake, which has high-quality water resources and pasture resources that provide an alternative ecological source site for large herbivores. In addition, the ecological source sites are concentrated in the eastern part of Qumalai County, with few but clustered distribution sites. Xinghai County also plays an important role in protecting bharal ecological source sites by bridging the species’ migration through Maduo and Gonghe counties. The ecological corridor of bharal is shown in Figure 9. The corridors in Qumalai and Maduo counties frequently interact. The northern part of Xinghai County is the main pathway for interaction among various ecological source sites. Therefore, protection measures should be launched. Furthermore, there may be more bharal migration paths between Republican and Maduo counties that are not found due to the limitation of the study area. Nevertheless, only some parts of the Sanjiangyuan District were selected to facilitate the reference for better measures to protect the species’ potential distribution in the Sanjiangyuan area.

DR was taken as a human disturbance factor reflecting the impact of human activities on current and future suitable habitats. However, the result showed that the bharal highly responded to the DR. Guo XJ [55] studied the number and distribution of bharal in Maduo County, and it was proven that bharal preferred areas within 1 km of rural settlements. Oli [56] discussed the selection of winter resting places for bharal in the Helan Mountains and found that such species preferred areas with less than 500 m of anthropogenic disturbance, which is generally consistent with our study. The potential habitat is mainly influenced by precipitation, isothermality, and LUCC. Bharal’s way of avoiding predators is mainly related to their climbing talent in steep areas, making them prefer suitable natural conditions in habitat selection. The roads around the habitat have little influence on them. Therefore, to establish species nature reserves and ecological corridors of bharal and other analogous species, such as the Tibetan gazelle (*Procapra picticaudata*), the main focus should be on the constraints of natural geographic factors. With global climate change, the species’ habitat is gradually lost. Hence, establishing an ecological corridor for bharal in the Sanjiangyuan District will help to protect the reproduction and survival of the population. The following recommendations were proposed for the optimization of the habitat suitability area for bharal in the Sanjiangyuan District of Qinghai Province:

(1) Human activities not only directly affect the diversity and distribution of the species but also influence the survival and development of the species through climate indirectly [57]. Care should be taken to avoid direct and strong impacts of human activities on species to conserve diversity, and the pollution and emissions on the overall climate should be addressed.

(2) The distribution range of suitable habitat for bharal is mainly related to climatic and geographic factors, such as precipitation, isothermality, distance to high-coverage grassland, and distance to water. Therefore, an appropriate geographic environment suitable for animal survival, residence, and development should be established to establish a bharal habitat protection zone. Additionally, protecting the ecological security of the habitat is insufficient. It is also vital to provide a guarantee for the survival and development of the potential habitat where the centers would be transferred.

(3) A four-county ecological corridor for bharal across eastern Qumalai, Maduo, Xinghai, and Gonghe counties should be established. The degradation of pastures and the emergence of climate extremes caused by excessive development should be followed in a timely manner. The establishment of the ecological corridor is conducive to guaranteeing survival, reproduction, and migration activities and contributing to the conservation of species diversity in the Sanjiangyuan District.

(4) Bharal are large herbivores, and the quality and quantity of grassland have a significant impact on their survival and development. In fact, the continuous degradation of grassland ecosystems in the Qinghai–Tibet Plateau region has attracted great attention from various parties and has become a major issue in grassland and restoration ecologies [58]. Long-term overgrazing is deemed a crucial human activity factor leading to the degradation of alpine grasslands. Therefore, the relative population between domestic animals and wildlife should be maintained to avoid further sharpening the conflicts [59].

### 4.3. Innovations, Limitations, and Prospects

The innovation of this study mainly lies in the adoption of the coupled MaxEnt-MCR model, which uses the contribution rate and resistance range of each environmental factor derived from the MaxEnt model, performing superior to some subjective methods, such as an expert scoring method and the Analytic Hierarchy Process [60,61]. This new model can effectively and scientifically provide support for predicting species’ potential distribution and establishing ecological corridors to help species survive, reproduce, and migrate. At present, there is no research using the coupled MaxEnt-MCR model, and the construction of ecological corridors for bharal has not yet been reported. MaxEnt-MCR model can make up for this lack and provide support for the species distribution modelling as well as the construction of ecological corridors. In addition, the UAV survey was implemented in this study to acquire the precise distribution points of bharal, which is helpful in improving the accuracy of species distribution modelling. Furthermore, the CA–Markov model was employed to predict the future LUCC under different climate scenarios, achieving a comprehensive simulation of future scenarios. The analysis and prediction of the temporal and spatial evolution of potential habitats and the changes in landscape patterns under climate fluctuations are helpful for the conservation of species diversity in the Sanjiangyuan District.

However, the deficiencies of this work are expounded as follows: (1) The bharal occurrence points obtained from the UAV survey were concentrated in Maduo County because it was found that the bharal were mainly distributed with Maduo County through household surveys and interviews for herders in the early stage. However, using UAV for the survey, we found that the small loading capacity of UAV makes it difficult to integrate multiple sensors on the same platform for observation. Limited by the battery capacity, the UAV has a very short flight time. Therefore, it is necessary to replace them with crewed aerial vehicles than UAVs, which can cover larger distances and load more weight. (2) The time series of all environmental factors were not realized in the compilation of future environmental factors. The simulation for future DEM, NPP, and railroad data was not available due to the limitation of the data set. (3) When the environmental variables were selected, the influence of extreme weather and natural disasters on the species distribution modelling was not considered.

In the future, it is intended to employ UAVs to verify and update the occurrence sites of bharal with GBIF on a regional scale to supplement the bharal database. During the species distribution modeling, extreme weather and disasters that are devastating to species can be investigated and taken as environmental factors.

## 5. Conclusions

In this study, the species occurrence points were integrated both from the GBIF dataset and UAV survey points and the potential habitat distribution of bharal in the Sanjiangyuan District was simulated. The MaxEnt model revealed and predicted the potential suitable habitat distribution of bharal in the Sanjiangyuan District under different climate scenarios.

The result shows that the calculated AUC value was 0.995 ± 0.004 (mean ± SD), and the simulation results were extremely excellent and similar to the distribution of sampling sites, indicating that the MaxEnt model can well-model species distribution. The results showed that the main environmental factors affecting the distribution of habitat were Bio16, Bio15, Bio3, DR, DH, and LUCC. The most suitable area for bharal should have less precipitation, better isothermal, and be close to high-coverage grassland and roads.

Under different shared socioeconomic pathways, the climate has changed, and LUCC will be degraded. A CA–Markov model was utilized to simulate the future LUCC under different climate scenarios by modifying the land transfer matrix. The climate scenarios of SSP-245 and SSP-585 were selected to perform multiple predictions of future environmental factors and scientifically predict the distribution of future bharal habitats.

At present, the total area of suitable habitat for bharal is 4669 km^2^, and the area of the most suitable habitat is 1237 km^2^, which is spatially distributed in Maduo and Gonghe counties and a few in Qumalai County. Under future climatic scenarios, the habitat is decreasing. Under the SSP-245 climate scenario, the total suitable area of the bharal habitat decreases to 3470 km^2^ in 2030 but is relatively good in 2050. Under the climate scenario of SSP-585, the total area of suitable habitat for bharal in 2030 drastically decreases by 44.6%, The total area of suitable habitat for bharal in 2050 decreases by 48.4%, and there is basically no suitable habitat for bharal in Republican County.

The MaxEnt and MCR models were coupled by using the contribution rate of each environmental factor in the MaxEnt model as the weight of the MCR model and using the contribution threshold of each environmental factor as the basis of the resistance value of the MCR model. The minimum cumulative resistance surface was calculated, and the ecological corridor of the suitable area for the bharal habitat was analyzed using the MaxEnt-MCR model, which provides a scientific basis for the migration and reproduction of bharal. The establishment of ecological corridors for bharal provides a realistic reference for the construction of the natural reserve. It is conducive to formulating measures to protect suitable habitat areas for bharal and other similar species.

## Figures and Tables

**Figure 1 ijerph-19-17082-f001:**
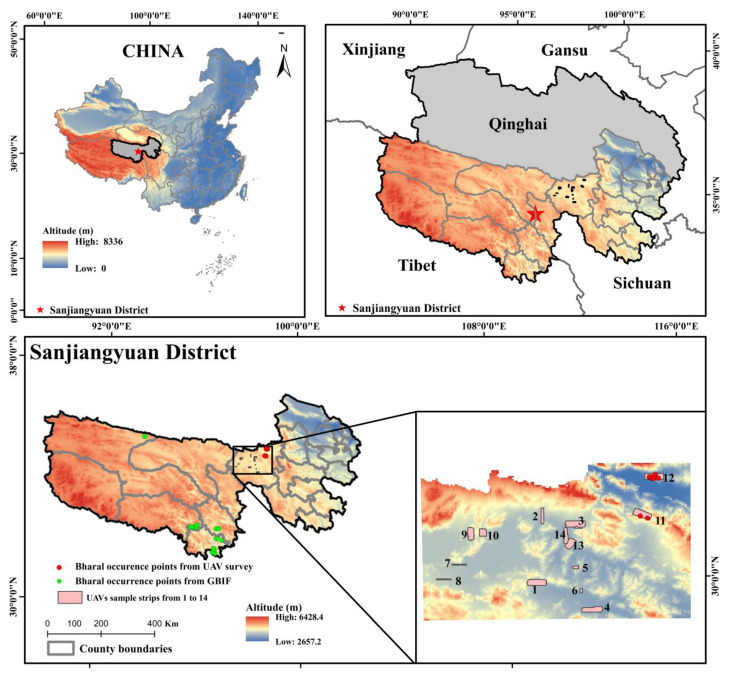
Study area and bharal occurrence points.

**Figure 2 ijerph-19-17082-f002:**
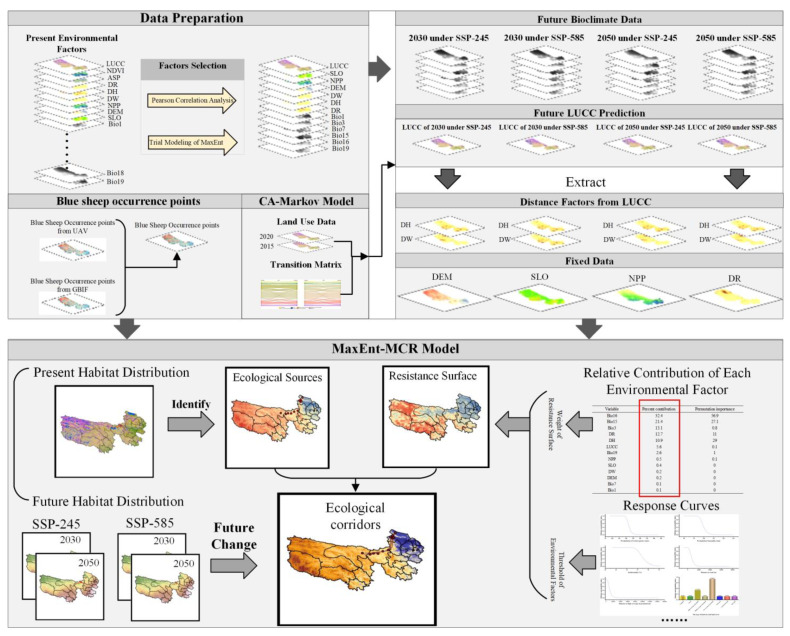
The workflow chart of the present research.

**Figure 3 ijerph-19-17082-f003:**
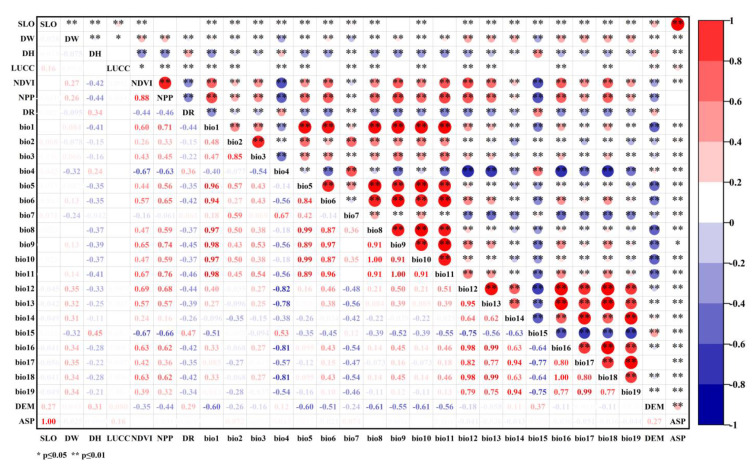
The multicollinearity of all environmental factors.

**Figure 4 ijerph-19-17082-f004:**
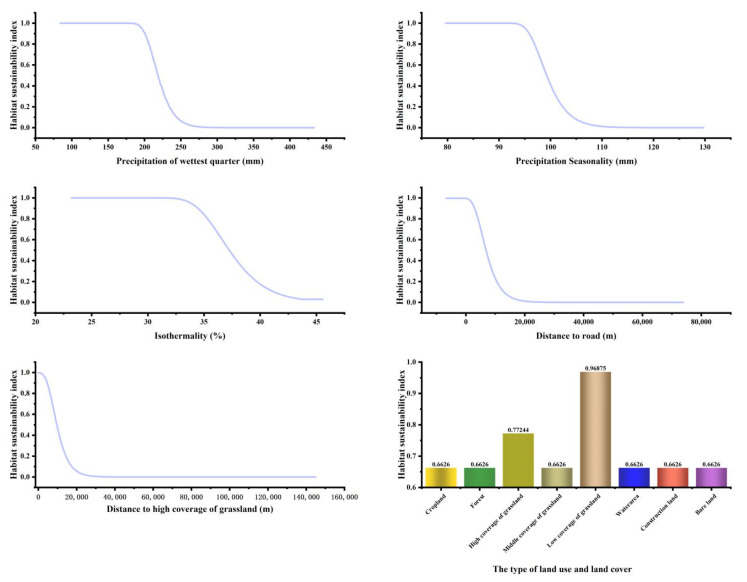
The response curve of main factors.

**Figure 5 ijerph-19-17082-f005:**
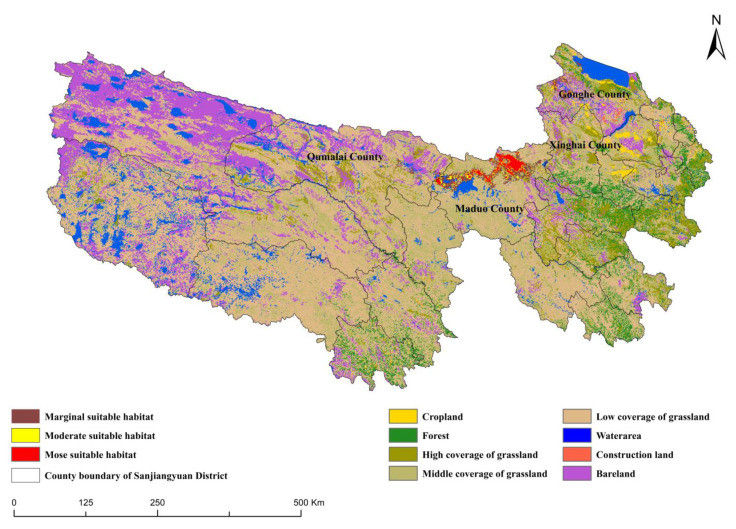
The suitable habitat area in 2020.

**Figure 6 ijerph-19-17082-f006:**
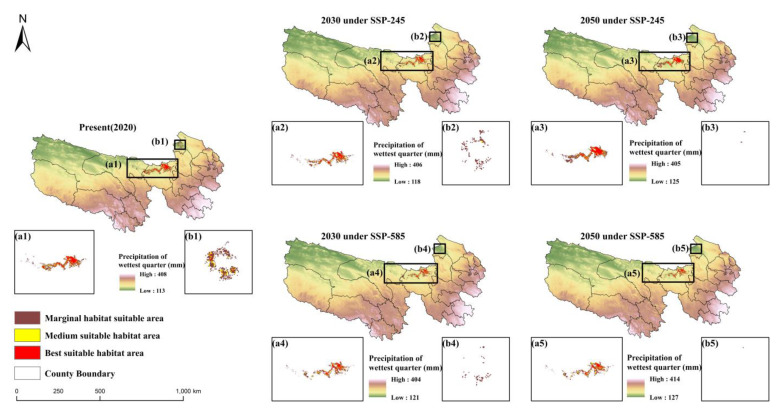
The spatiotemporal evolution of suitable habitat under future climate change. The mainly differences of the suitable habitat have been shown in the subfigure **a1**–**5** and **b1**–**5** respectively.

**Figure 7 ijerph-19-17082-f007:**
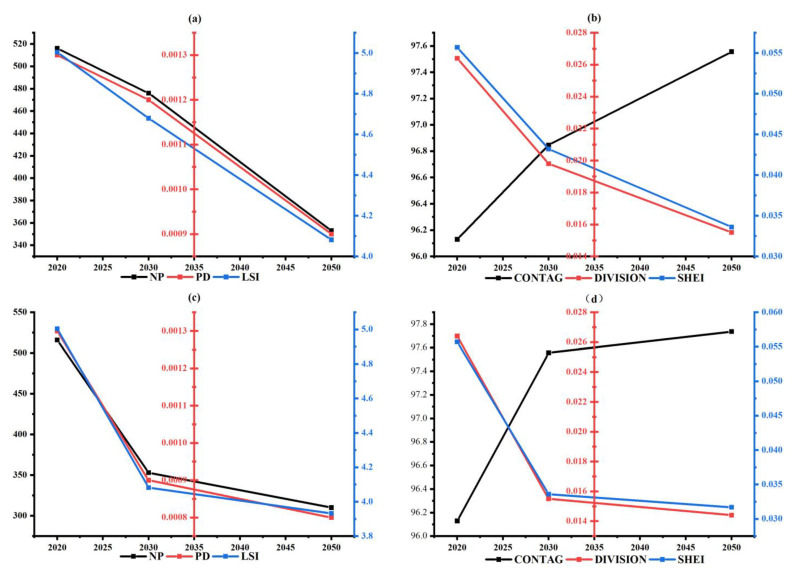
The landscape pattern index evolution in the Sanjiangyuan District; (**a**,**b**) are the changes of the landscape pattern index under SSP-245; (**c**,**d**) are the changes of the landscape pattern index under SSP-585.

**Figure 8 ijerph-19-17082-f008:**
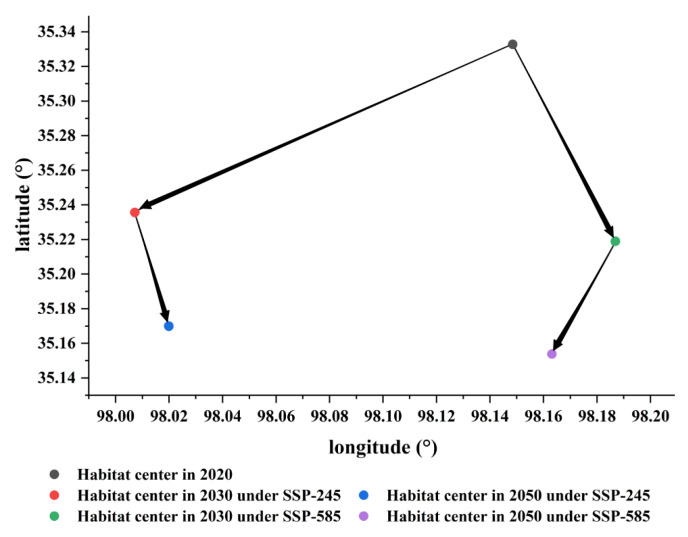
The transformation of habitat center.

**Figure 9 ijerph-19-17082-f009:**
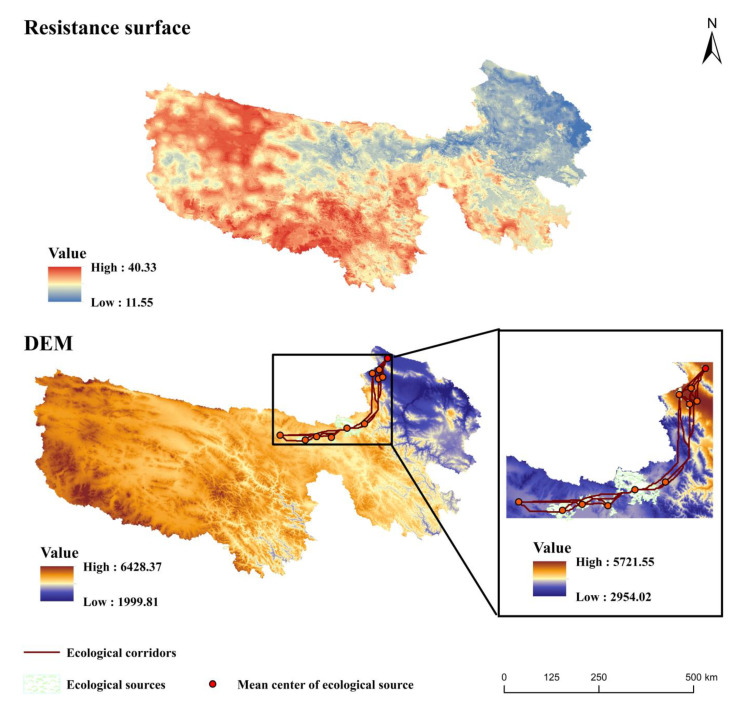
The resistance surface and ecological corridors of the Sanjiangyuan District.

**Table 1 ijerph-19-17082-t001:** The parameters of UAVs in this research.

Parameters	Electric Fixed-Wing UAV	F1000 Electric Fixed-Wing UAV
Wingspan	1.6 m	1.6 m
Payload	0.5 kg	1 kg
Maximum take-off weight	3 kg	3 kg
Engine	Electric	Electric
Endurance time	90 min	60 min
Flight speed	72 km/h	60 km/h
Camera model	ILCE-5100	ILCE-5100
Number of integrated cameras	2	1
Focal length	30 mm	30 mm
Pixel size	6000 × 4000	6000 × 4000

**Table 2 ijerph-19-17082-t002:** The resistance value and weight of factors.

Resistance Factors	Resistance Value					Weight
	10	20	30	40	50	
LUCC	High coverage of grassland	Low coverage of grassland	Cropland, forest, middle coverage of grassland, bareland	Water area	Construction land	32.4
Bio1	[−16, −6]	(−6, −4]	(−4, 2]	(−2, 1]	(1, 8]	21.4
Bio3	[−25, 32]	(32, 35]	(35, 37]	(37, 39]	(39, 45]	13.1
Bo7	[36, 38]	(34, 36]	(38, 40]	(30, 34]	(40, 43]	12.7
Bio15	[80, 95]	(95, 100]	(100, 105]	(105, 110]	(110, 130]	10.9
Bio16	[110, 180]	(180, 220]	(220, 235]	(235, 250]	(250, 408]	5.6
Bio19	[2, 8]	(8, 12]	(12, 16]	(16, 21]	(21, 36]	2.6
DW	[0, 5000]	(5000, 10,000]	(10,000, 15,000]	(15,000, 20,000]	(20,000, 50,000]	0.5
DH	[0, 5000]	(5000, 10,000]	(10,000, 15,000]	(15,000, 20,000]	(20,000, 50,000]	0.4
DR	[0, 5000]	(5000, 10,000]	(10,000, 15,000]	(15,000, 20,000]	(20,000, 70,000]	0.2
SLO	[0, 3]	(3, 9]	(9, 15]	(15, 21]	(21, 26]	0.2
NPP	[0, 80]	(80, 170]	(170, 280]	(280, 528]	(528, 615]	0.1
DEM	[4500, 5000]	(4000, 4500]	(3500, 4000]	(5000, 6430]	(2000, 3500]	0.1

**Table 3 ijerph-19-17082-t003:** The area of suitable habitat under future climate change.

Category (km^2^)	Present (2020)	2030		2050	
		SSP-245	SSP-585	SSP-245	SSP-585
Marginally suitable habitat area	2668	2059	1555	1883	1428
Moderately suitable habitat area	764	558	450	467	450
Most suitable habitat area	1237	853	581	1035	529
Total area of suitable habitat	4669	3470	2586	3385	2407

## Data Availability

The data that support the findings of this study are available from the corresponding author (F.Y.) upon justifiable request.

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
