# Peer review of "Influences of Climate Change and Land Use Change on the Habitat Suitability of Bharal in the Sanjiangyuan District, China"

_ijerph, 2022, doi:10.3390/ijerph192417082_

Round 1
Reviewer 1 Report
Using Sanjiangyuan District as a study site, this study used an unmanned aerial vehicle (UAV) was used to investigate the number and location of bharal (Pseudois nayaur). They simulates the future geographical distribution of bharal habitat under different climate scenarios. They predicted a complete loss of habitat in Gonghe County by 2050, revealing the landscape in Sanjiangyuan District will be more fragmented and complex.
Abstract: The abstract was a good summary of what the paper concluded.
Introduction: The introduction is well written with appropriate literature review.
Methods: It seemed as though the survey design was not adequately presented. I don't know how the authors used the UAV to survey their 14 sites. There is really no information about the UAV surveys. I'm suggesting major changes to this section, including the addition of a figure of where the 14 sites were and how they were flown. You have listed "UAV sample route" in the legend, but I cannot see them if they were there. It looks like the waterway rivers are too large on the map. Since the main novelty of the study is using UAV collected data, then this section is very important. What type of UAV did you use? What were the flight parameters, height, altitude, and speed of flights? What type of flight method did you use. Did you use transects? How big were the areas that you flew? How did you process the data finally for detections of wildlife?
I'm a bit concerned that the study area of the entire Sanjiangyuan District, China was a bit too ambitious, unless you adequately sampled this entire area, then why did you use it for the scale of the study? You included GBIF points that also seem clustered into one area. Is there any indication of where those data came from or how comprehensive the district has been sampled? You mention in the limitations discussion that "UAV survey used in this paper are mainly concentrated in Maduo County" and also "In the future, we can use UAV to survey the entire Sanjiangyuan District to avoid the problem of small sampling."
It seems that your study should only consider counties or areas where data were actively collected, or surveys were completed. Otherwise, you do not really know that the other areas are absent of bharal. There seems to be no evidence that the surveys were comprehensive over the entire district, and are merely opportunistic sampling which can bias the results and models. There is also no discussion of spatial autocorrelation, or how you may have resolved potential autocorrelation in the model.
Maxent in this case is using all of the presence only data that you collected, and from GBIF. However, the scale at which you projected your models is huge, and well beyond your survey regions.
The reason I'm bringing this up as being so critical, is also because the GBIF presence points do not wind up being considered suitable habitat in your Maxent maps. Why is there no suitable habitat where all of the GBIF points were found? If the model was 99% accurate for modeling bharal habitat then we would have expected to see some suitable habitat where the bharal were found in the GBIF data. Your model basically says that entire area from the GBIF data is not suitable for bharal. Finally, in the discussion I see one line "This research used UAV survey points combined with MaxEnt model to simulate the 445 potential habitat distribution of bharal in the Sanjiangyuan District" Did you even use the GBIF data?
Your discussion mentions "The results showed that the suitable habitat of bharal was mainly distributed in Maduo County, which is the source area of the Yellow River" Since this was the only county you surveyed then it seems you biased your entire study this way. Perhaps you should subset your study only to Maduo county where you sampled instead.
You make broad statements:
"Analyzing the contribution range of environmental factors based on MaxEnt-MCR model, we found that bharal preferred ranges closer to roads."
Although it may be that your samples were all collected next to roads, and that you biased your study. Since we have no information about how you collected the UAV data or whether there was a fair distribution of these distances from roads, then it seems you could have potentially only collected data near roads and then drawn this conclusion.
The modeling is really interesting overall, with the future climate scenarios. Although this part was also confusing to me because you seem to have a LULC simulation in this paper. It almost seems as though this information could go into another paper about simulating LULC, or go into an appendix and referenced. This section is very confusing because you mention: "Under the modern climate, the total area of habitat in the Sanjiangyuan District is 225 about 4669 km² (see Figure 5)" but figure 5 is merely a LULC map. As a reader, I have no idea how you determined suitable habitat from these LULC maps as they are not habitat suitability maps. There is no information about what a,b,c,d even represent, and there is no indication of what is suitable bharal habitat. Fig 5, 6, and 7 are all unnecessary LULC projections that could go into another standalone manuscript or be put into an appendix and referenced in a paragraph in this manuscript.
Once we get to Fig 8 then we can see the suitable maxent produced habitat maps.
It really shows that you have expertise in modeling and producing good results. The connectivity modeling also was interesting.
Overall, due to the issues I mentioned above, this manuscript shows serious flaws that I would not recommend it for publication. You could subset the entire study to the areas where you surveyed in Maduo county and perhaps a small scale of regional projection would make sense. Also, the LULC mapping needs to be reformatted and either put into a Supplement, Appendix, or into another manuscript entirely. Then you could reference those supplements in this manuscript, and present only information relevant to the bharal distribution and mapping.
Author Response
Dear Reviewer:
Thank you for your valuable comments, which are much important to the ariticle. Please see the attachment.

Reviewer 2 Report
This paper studied the suitable habitat of Yan sheep, a unique herbivorous wild animal in the Qinghai-Tibet Plateau, and discussed its future changes, revealing the potential threat of climate and land use change to wildlife. It is a significant study. However, english writing and grammar need improvement.
Line 14 “The results showed that…was…”, this sentence should be in the past tense.
Line 34-42 It is suggested to switch line34-38 and line39-42. Firstly, climate change has a great impact on biodiversity, and then the Qinghai-Tibet Plateau is particularly sensitive to climate change due to its special environment, leading to greater threats to wildlife living on it. This may make the paragraph logic clearer.
Line57 Animal escapes is one of the reasons of duplicate counts, so the sentence should be “the line intercept method is prone to duplicate counts due to animal escapes and the influence of environment…”
Section 2.2.1 Please show the bharal occurrence sites in a figure.
Figure 5 The figure shows a distribution of lucc, not the suitable habitat you described.
Figure 8 A more appropriate figure name is needed to accurately describe what the image contains.
Author Response
Dear Reviewer:
Thank you for your valuable comments.
Please see the attachment.

Round 2
Reviewer 1 Report
The new draft has been majorly improved. However, there are still minor changes that need to be made.
Line 8 and 9, the word whose is grammatically incorrect because climate is not a person, and also habitat shape is incorrect and normally is habitat structure.
Please use this line for the first two lines of the abstract instead:
"One of the biggest dangers to the degradation of biodiversity worldwide is climate change. Its oscillations in the future could result in potential alterations to species populations and habitat structure"
Line 10, the word taken should be deleted
Line 62- line intercept I think should be line intercept transect instead
Line 67, you don't need to capitalize Bharal it should be bharal unless you are using a scientific name then the species name is not capitalized.
Line 122- GBIF data had how many presence points? If they are green on the map why are there no regions in this area that were identified in Maxent. I think you should add a section in the discussion about why the GBIF presence points and the distribution map does not agree. Otherwise, your model does not make much sense, this is a comment I made in the last draft revision and I do not see it addressed anywhere.
Line 186: What is a sortie?
Line 318 Is it really Republican county?
Line 407- this first paragraph of the discussion is really general and seems to not add much to the manuscript. I would delete most of this and discuss the findings of your research. One or two general sentences is fine, but this paragraph is too long.
Line 485 what are steep bluey areas? I don't know the word bluey maybe delete it.
Line 553 what is resting time? The UAV has a short resting time? What does that mean?
Line 554-555. The rationale here does not make sense. Can you use remote sensing to detect bharal? I don't think so, even with high resolution remote sensing this would not be very easy. Why are UAV surveys not good for long term wildlife monitoring? Perhaps instead you might suggest manned aerial flights which can cover larger distances.
Line 571 - you still used UAV and GBIF data together right? so move lines 576 up to line 571 and improve it.
The paper is significantly improved for writing and content. I appreciate you integrating my comments previously. I would still consider rewriting some of the manuscript for clarity. I did not go through
Author Response

(The authors gave the same response as above.)
